# Relationship between Principals’ Leadership Styles and Teachers’ Behavior

**DOI:** 10.3390/bs13020111

**Published:** 2023-01-28

**Authors:** Kazi Enamul Hoque, Zarin Tasnim Raya

**Affiliations:** 1Faculty of Education, University of Malaya, Kuala Lumpur 50603, Malaysia; 2Independence Researcher, Kuala Lumpur 50603, Malaysia

**Keywords:** instructional leadership style, democratic leadership style, transformational leadership style, laissez-faire leadership, teacher behavior

## Abstract

An effective leader follows a style that helps maintain good relations with his staff. A school leader should use a style best suited to his teachers’ behavior. This research investigates the association between four leadership styles (instructional, democratic, transformational, and laissez-faire) and teachers’ behavior in Malaysia. This study applied a quantitative research method using a survey technique by administering questionnaires. Both descriptive and inferential statistics were used to analyze the data. Multiple regression and ANOVA were used to study the strength of the relationship between variables. The research found average care of the principals towards teachers’ emotional behavior. Democratic leadership style showed significant relationships that explain 28.5% of the variation in the emotional behavior of the teachers. Leaders with a democratic leadership style were more aware of and responded positively to teachers’ psychometric behavior. School principals with transformational leadership styles responded positively to teachers’ pro-social behavior, although the relationship was weak. These results indicate that a democratic leadership style addresses the issues of teachers’ emotional behavior, while instructional leadership, which is the most perceived leadership style, does not. The result of this study can guide Malaysian school principals in choosing the appropriate leadership style best suited to teachers’ behavior.

## 1. Background of the Study

Leadership has been defined as the ability to steer a group towards a shared goal that would otherwise not be met in the leader’s absence [1]. This ability may be explained as the style in which the leader behaves [2] with the people they lead to attain the group objectives. For example, in the educational setting, studies exploring the perception of principals [3,4] and teachers [5,6,7] suggest that school leaders adopt various leadership styles, including autocratic, bureaucratic, democratic, instructional, transformational, transactional, moral, democratic, or laissez-faire leadership to achieve educational or organizational objectives.

Ali (2017) [8] argued that the appropriate leadership style would depend on the school’s context and the maturity of the staff and, in practical terms, will require the school leader to adopt several leadership styles or change their style when the situation demands. This view has been corroborated in several empirical studies [9,10]. For instance, Chen et al. [9] recently showed that principals in Germany demonstrating instructional and integrated leadership had higher student achievements, while principals demonstrating transformational leadership in China had higher student achievements.

Nonetheless, school leadership that takes a people-first approach may be crucial in driving a positive school culture conducive to better learners’ educational outcomes [11]. Several meta-analytic studies have shown that educational leadership styles—distributive and transformational leadership styles in particular—are associated with student achievement [12,13]. Additionally, a survey of Indonesian teachers suggests that school principal’s transformational leadership may be associated with a rational decision-making style [14,15], which may have important implications for fostering a positive school culture and organizational image [5,16,17,18]. A principal’s leadership style (PLS) is even more important in determining educational outcomes and school culture in challenging settings [6,10,19].

In addition to student-related and organizational outcomes, there is robust evidence for the association between PLS and teacher-related education or non-educational outcomes. For instance, Pietsch and Tulowitzki [20] showed that the PLS has both direct and indirect impacts on teachers’ instructional practices, including classroom management, student orientation, and enhanced activities, as well as their work setting, innovation capacity, and motivation. Others have shown that teacher-perceived PLS affects teacher’s self-efficacy, work performance, motivation, job satisfaction, and organizational commitment, with positive outcomes associated with transformational leadership style [15,21,22,23,24,25,26,27,28,29,30].

Although the association of student-, teacher- and organization-related outcomes with the PLS is well-documented, the impact of leadership style on teachers’ emotional, pro-social, and psychometric needs has been sparsely reported. Teachers’ emotional conduct has been proven to be significantly influenced by leadership style. Different leadership philosophies, including transformational, transactional, and autocratic, can generate various emotional reactions in teachers, according to academic research. For instance, it has been discovered that transformational leadership increases teachers’ levels of job satisfaction, motivation, and dedication [22,23,24]. Autocratic leadership, on the other hand, has been associated with poorer levels of work satisfaction and higher levels of stress among teachers [15,25]. Additionally, different leadership styles might affect teachers’ emotional well-being, burnout, and turnover differently [29,30]. Therefore, this study aims to examine how four PLS (instructional, democratic, transformational, and laissez-faire) affect teachers’ emotional, pro-social, and psychometric needs from the perspective of teachers.

## 2. Conceptual Framework

Principals, as school leaders, generally follow either instructional [31,32], democratic [31,32], transformational [33], laissez-faire leadership styles (Lewin et al., 1939) [34], or a combination of them. The suitability of the style depends on staff behavior. An effective leader (either a principal or any other organizational leader) must use the right style to achieve organizational goals. Thus, leadership is a social influencing process in which a leader seeks the voluntary participation of subordinates to reach organizational goals [35].

According to Leithwood, and Leithwood and Jantzi [36,37], instructional leaders develop, direct, and supervise the curriculum and instructions. This leadership style establishes high expectations among teachers and students. It indicates that instructional leaders are aware of teachers’ pro-social (high expectations towards students) and psychometric (cognitive abilities such as curriculum and instructions) behavior such as cognitive abilities (curriculum and instructions) [38,39]. An instructional leadership style, which focuses on content, instruction, and evaluation, has a good effect on teachers’ emotional and psychological requirements. According to research, teachers who are supported by instructional leadership are more likely to express higher levels of job satisfaction, motivation, and dedication, as well as lower levels of stress and burnout [40]. Instructional leaders frequently concentrate on developing a solid curriculum and giving instructors the tools and assistance they require to give high-quality education. Teachers may feel that their efforts are having an impact on their students’ lives, which can give them a feeling of purpose and fulfillment [41].

Similarly, democratic leaders prioritize group interest by practicing social equality [31,32], which indicates the awareness of emotional behavior toward staff. It has been discovered that a democratic leadership style that emphasizes cooperation and shared decision-making, as well as a participatory approach has a favorable effect on teachers’ emotional and psychological requirements [42]. According to studies, teachers who work in environments with democratic leadership are more likely to express better levels of job dedication and satisfaction as well as lower levels of stress and burnout [43]. Democratic leaders frequently provide teachers the chance to express their thoughts and ideas, and they frequently involve their followers in decision-making and goal-setting. Teachers may feel empowered, included, and that their efforts are valued as a result of this. They may also feel like they are making a difference. Additionally, democratic leadership may create a climate at work that is constructive and helpful that addresses the particular [44].

Transformational leaders also offer vision and purpose to emotionally motivate the staff to do more than intended [33]. Managers with a transformational leadership style concentrate on the growth and development of the value system of employees, their inspirational level, and moralities with the preamble of their abilities [45,46]. On the other hand, laissez-faire leaders give the staff the power to make their own decisions [34], indicating awareness of the staff’s emotions [47,48]. It has been discovered that the laissez-faire leadership style, which is defined by a hands-off attitude and a lack of direction or supervision from the leader, negatively affects teachers’ emotional and psychological requirements. Teachers who experience this kind of leadership may feel unsettled, perplexed, and unmotivated. According to studies, teachers who work in environments with lax leadership are more likely to experience stress and burnout, as well as lower levels of job dedication and satisfaction [49,50].

The aforementioned discussion suggests that principals’ leadership ideologies can have a big impact on the emotional, pro-social, and psychometric requirements of teachers. A good and encouraging work environment that supports the emotional and pro-social needs of teachers can be created, for instance, through a transformational leadership style that focuses on inspiring and motivating teachers to work toward a common vision. The autocratic leadership style, on the other hand, which is based on rigorous rules and demands, might produce a stressful and unsupportive work atmosphere that might not suit the emotional or pro-social requirements of teachers. The degree of autonomy and professional development opportunities that teachers have may also be influenced by the leadership style of a principal, which may have an impact on their psychometric requirements.

Thus, this study’s conceptual framework was developed (Figure 1) to investigate how instructional, democratic, transformational, and laissez-faire leadership styles relate to teachers’ emotional, pro-social, and psychometric needs. Within the scope of this aim, the following research questions were explored from the teachers’ perspective:

What degree of leadership style practice do Malaysian school leaders perceive among principals?

Which leadership style do teachers prefer?

What are the perceptions of the teachers on the principal’s awareness and handling of their emotional, pro-social, and psychometric needs?

Is there a significant relationship between the teachers’ perceptions of PLS and their emotional, pro-social, and psychometric needs?

## 3. Methods

### 3.1. Ethical Statement

The study protocol was approved by the University of Malaya Research Ethics Committee (approval number: um.863/hru.op.rf). The permission from respective school principals was obtained to access the teachers at respective schools for data collection after explaining the research purpose. All participants were explained that the questionnaire was strictly anonymous, there was no need to disclose any sensitive identity in the questionnaire, and that strict confidentiality would be maintained while handling data. The respondents were asked to read the instructions and were free to withdraw at any moment if they felt the questions were offensive, threatening, or embarrassing.

### 3.2. Research Design and Study Participants

Research has been divided into the following two main groups according to methodology: (1) deduction and (2) induction. Comparatively, the fundamental methodologies to study are divided into the following three categories: (1) quantitative; (2) qualitative; (3) mixed methods [51]. According to Muijs [52], the use of quantifiable tests in a logical approach to study a preset notion or hypothesis is referred to as “a quantitative research technique; positivism.” In order to ascertain the degree of relationship among the study’s variables, this study employed a quantitative technique using deductive reasoning. The main objective of the study was to examine the level of practice for principals’ leadership styles, finding the most preferred leadership style and the teacher’s perception of the principal’s awareness and handling of their emotional, pro-social, and psychometric needs at Malaysian schools. This research also attempted to look into the relationships between study variables. Thus, this study utilized a survey technique. School teachers (*n* = 258) from government-run primary schools in Selangor state, Malaysia, willing to participate in the survey were chosen randomly to respond to questionnaires distributed through face-to-face meetings, emails, or an online form shared via WhatsApp messaging application. The survey was conducted between 2020 and 2021.

### 3.3. Instrumentation

A self-administered and adapted three-part questionnaire was used as the primary data collection instrument. Part A (self-administered) included the demographics of the respondents, such as gender, ethnicity, education, and work experience.

Part B (adapted) surveyed leadership practices based on the Principals Instructional Management Rating Scale (PIMRS) [53], comprising 32 items. The following four leadership styles were measured: instructional leadership (three items, e.g., I am happy if my Headmaster/Principal shares the school’s vision and mission with the school community), democratic leadership (four items, e.g., I am happy if my Headmaster/Principal encourages teachers to accomplish a task that directed), transformational leadership (four items, e.g., I am happy if my Headmaster/Principal is open-minded in extracting and giving feedbacks), and laissez-faire leadership (four items, e.g., I am happy if my Headmaster/Principal gives subordinates complete freedom to solve problems on their own).

Finally, part C (self-administered) surveyed the teacher-perceived principal’s handling of their emotions (four items, e.g., My principal cares if I feel helpless), pro-social (five items, e.g., My principal appreciates if I often help people without being asked), and psychometric (nine items, e.g., My principal cares if I have difficulties falling asleep) behavior.

### 3.4. Data Analysis

Data were analyzed using the Statistical Package for Social Sciences (SPSS version 24; IBM Corp., Armonk, NY, USA) software. Survey items in part B and part C were measured using a five-point scale (1 = strongly disagree to 5 = strongly agree). Both descriptive and inferential statistical methods were used to analyze the data. Demographic characteristics of teachers (gender, education level, ethnicity, and work experience) were assessed using frequency and percentage. 

The arithmetic means and standard deviations (SD) were interpreted for teacher-perceived leadership styles and principal’s handling of their emotional, pro-social, and psychometric behavior as follows: mean value 1-3 was considered ‘low perception,’ a mean value between 3-3.99 was considered ‘average perception,’ and a mean 4-5 was considered ‘high perception’ [54]. The relationship between teacher-perceived PLS and the principal’s handling of teachers’ emotional, pro-social, and psychometric needs was investigated using multiple regression and ANOVA.

Before analysis, the data were checked to identify outliers and missing values. A Box plot was used to detect the outliers. Hair et al. [55] suggested that if extreme marginal data out of the total is found, the respondent needs to be excluded from the analysis. Thus, respondents who had extreme marginal data were excluded from the study. After cleaning the data from 258 teachers who responded to our invitation to participate, 213 responses were included in the analysis. After frequency analysis among these 213 respondents, no missing data were identified. Several modalities were used to assess normality, including coefficient of peak mode, median, and mean; variables that have mean, mode, and median matching; skewness and kurtosis. The skewness value for every item was <2 and within the range of +2 to −2, indicating normal data distribution.

Since the survey items for leadership styles and teachers’ behavior in this study were adapted from PIMRS [53] and self-administered, an exploratory factor analysis (EFA) was used to determine their validity and the suitability or cohesion with the covariance structure of the measured variables. Sampling adequacy was assessed using the Kaiser-Meyer-Olkin (KMO) test and Bartlett’s test of sphericity. The KMO value of 0.722 (KMO < 0.900) indicated that the data were suitable for factor analysis and a significant approximate chi-square in Bartlett’s test of sphericity (χ^2^ = 4127.172, *p* < 0.0001) indicated high data factorability.

The result from the pilot study confirmed that the items in both questionnaires were relevant, although some minor changes were required. To verify the convergence validity of the instrument used for this study, multi-item scales were analyzed based on factor analysis. The scales include four predictor variables, such as instructional leadership, democratic leadership, transformational leadership, and laissez-faire leadership, and study three criterion variables, such as teachers’ emotional behavior, their pro-social behavior, and their psychometric behavior. Underlying assumptions were observed before proceeding to the subsequent phases of factor analysis. The dimensionality of principals’ leadership styles and teachers’ behavior was also determined using exploratory factors and reliability analysis. The results of factor analysis for principals’ leadership styles came up with four factors with factor loading ranging from 0.60 to 0.87 while 0.58 to 0.91 for three factors of teachers’ behavior using principal component analysis and Varimax rotation procedures, amounting to 83.96% for PLS and 80.52% for teachers’ behavior of total variance (Table 1). In other words, all these items were internally consistent, all measuring the same basic construct. Thus, the common degree values corresponding to all the research items are greater than 0.4. Therefore, the corresponding relationship between the item and the research variable is basically the same as expected, which means that the research variables are effective.

Internal consistency reliability of all items was tested using the Cronbach alpha coefficient. The overall value of Cronbach’s Alpha coefficient for all scale items ranged from 0.72 to 0.82 for all answered items on the questionnaire. As shown in Table 1, Cronbach’s Alpha value for all study variables indicates adequate internal consistency and reliability.

## 4. Results

### Participants Profile

Overall, 18.8% of the respondents were male teachers, while 81.2% were female teachers (Table 2). The proportion of Malay, Chinese, and Indian teachers was 78.9%, 7.0%, and 10.8%, respectively (Table 1), representing Malaysia’s three main ethnic groups. Regarding teachers’ educational background, 58.8% had a bachelor’s, 38.4% had a master’s, and 0.5% had Ph.D. degrees. Additionally, 2.3% of the teachers had a diploma, which is becoming rarer because, for a decade, a bachelor’s degree has been the minimum qualification for teaching positions. Most participants (46.0%) had over 15 years of teaching experience (Table 2).

What degree of leadership style practice do Malaysian school leaders perceive among principals?

Teachers showed a high perception of all three items related to instructional PLS and suggested that principals documented guidelines (mean = 4.25 ± 0.522), shared their vision and mission (mean = 4.28 ± 0.432), and had sufficient curricular knowledge to instruct their teachers (mean = 4.70 ± 0.721) (Table 3). Similarly, teachers showed a high perception of the following four items used to evaluate the democratic PLS (Table 3): teachers agreed that principals foster friendly relationships (mean = 4.32 ± 0.412), gives opportunity to teachers (mean = 4.72 ± 0.701), support the teacher’s priority needs (mean = 4.18 ± 0.641), and encourage teachers to accomplish a directed task (mean = 4.22 ± 0373).

Teachers also highly perceived all four items related to the transformational PLS (Table 3). For example, teachers felt that principals: provided equipment and facilities (mean = 4.12 ± 0.711); were open-minded in extracting and giving feedback (mean = 4.78 ± 0.515); consistently made changes from the feedback that they were giving (mean = 4.68 ± 0.246); were relevant on current technological development and encourage strategic implementation (mean = 4.53 ± 0.321). However, teacher perception toward laissez-faire PLS was average for all four items (Table 3): Teachers indicated that principals: let subordinates work problems out on their own (mean = 3.80 ± 0.733), stay out of the way of subordinates as they do their work (mean = 3.23 ± 0.792), allows subordinated to appraise their own work (mean = 3.12 ± 1.141), and gives subordinates complete freedom to solve problems on their own (mean = 3.87 ± 1.013).

Which leadership style do teachers prefer?

Overall, teachers preferred the democratic PLS, followed by instructional, transformational, and laissez-faire leadership styles (Table 4).

What are the perceptions of the teachers on the principal’s awareness and handling of their emotional, pro-social, and psychometric needs?

Teachers had an average perception of the principal’s handling of their emotional needs on all four items (Table 5). Teachers indicated that the principal cared if they felt: left out of things (mean = 3.76 ± 0.711), lonely (mean = 3.67 ± 0.904), helpless (mean = 3.81 ± 0.366), or if they felt that they were someone else (mean = 3.73 ± 0.815).

In the following two out of five items, teachers had an average perception of the principal’s handling of their pro-social behavior: the principal appreciates if the teacher often does favors for others without being asked (mean = 3.78 ± 0.746), and the principal appreciates whenever the teacher lends things to people without being asked (mean = 3.71 ± 0.927) (Table 5). Teachers had a high perception for the following remaining three items: the principal appreciates whenever the teacher helps others without being asked (mean = 4.12 ± 0.836), the principal appreciates if the teacher often compliments people without being asked (mean = 4.22 ± 0.656), the principal appreciates if they often share things with people without being asked (mean = 4.20 ± 0.615).

Teachers had an average perception of the principal’s handling of their psychometric needs on all eight items (Table 5). They felt that the principal cared if they had the following: a headache (mean = 3.64 ± 0.718), stomach-ache (mean = 3.80 ± 0.710), backache (mean = 3.65 ± 0.705), or difficulties falling asleep (mean = 3.72 ± 0.838), and if they felt low (mean = 3.83 ± 0.760), irritable (mean = 3.45 ± 0.815), nervous (mean = 3.51 ± 0.757), or dizzy (mean = 3.62 ± 0.947).

Is there a significant relationship between the teachers’ perceptions of PLS and their emotional, pro-social, and psychometric needs?

In the multiple regression analysis, PLS was not associated with teachers’ emotional (R = 0.264; F = 1.712; *p* = 0.244) or psychometric (R = 0.355; F = 2.163; *p* = 0.081) needs (Table 6). Although PLS was significantly associated with the teacher’s pro-social needs, the association was weak (R = 0.302; F = 2.584; *p* = 0.023) (Table 6).

The multiple linear regression analysis findings corroborated with ANOVA (Table 7). The standardized coefficients of democratic PLS (*β* = 0.324, t = 2.584, *p* = 0.02) explain 32.4% of the variation in teachers’ emotional needs, which was significant.

However, there were no significant differences between other PLS (instructional, transformational, or laissez-faire) and teachers’ emotional needs. In addition, there was no significant difference between any leadership styles and the pro-social or psychometric needs of the teachers. However, both the regression coefficient for teachers’ pro-social and psychometric needs were positive.

## 5. Discussion

Earlier studies from Malaysia indicate that about 15% of school leaders practiced a mixed leadership style, while the majority used a specific leadership style [56]. However, the most effective leadership style in the school context was not evident from these prior studies. Interestingly, Wahab et al. [57] and Jones et al. [58] found that most school leaders in Malaysia prefer the transformational style and deem it effective in attaining better school and student outcomes than other PLS. Although transformational PLS was the second most perceived by teachers in this study after democratic style, there was a null association between transformational PLS and teachers’ emotional, pro-social, or psychometric needs. This finding contradicts the previous study by Ismail et al. [45], in which they described those managers with a transformational leadership style concentrating on the growth and development of the value system of employees, their inspirational level, and moralities with the preamble of their abilities. Further, this study observed no association between laissez-faire and instructional PLS and teachers’ emotional, pro-social, or psychometric needs.

In contrast, teachers in this study had a positive and high perception of the democratic PLS followed by instructional and transformational leadership styles. Furthermore, even in terms of the school leaders’ behavior towards the teachers’ emotional needs, the democratic PLS was more relevant, although there was no association between democratic PLS and teachers’ pro-social or psychometric needs. This result suggests that respondents see school leaders with a democratic leadership style as being more aware of and responding positively to their emotional behavior, which is consistent with earlier observations of Harris and Chapman [59], who showed that democratic PLS guides and empowers teachers in the school by distributing leadership responsibilities for the overall good of the school.

Similarly, Shepherd-Jones and Salisbury-Glennon [60] showed that teachers had better self-reported psychological needs scores for competence, followed by relatedness and autonomy, with principals with democratic leadership styles than those with authoritarian or laissez-faire leadership styles. Interestingly, school administrators participating in the same study, although they perceived their democratic leadership style, rarely reported supporting teachers’ competence as a motivational strategy [60]. These results underscore the significance of investigating the effect of leadership styles in the school environment from the perception of both the school leaders and the teachers they lead, as the perceptions are likely to vary.

In addition, a democratic PLS may positively impact teachers’ job satisfaction, as demonstrated by Nadarasa and Thuraisingam [61]. These findings are corroborated by Lopez Delgado [62], who reported that the democratic PLS facilitated improvements in schools through the collaboration and participation of teachers, students, and stakeholders. Moreover, evidence from Malaysia indicates the positive impact of instructional and distributed leadership (promoted under the Malaysia Education Blueprint policy) on student outcomes [63,64]. However, in line with previous studies [36,38], which described six themes of instructional leadership with no mention of emotional behavior, the current study suggests that instructional leadership may not adequately address teachers’ emotional needs.

More recent studies from Malaysia indicate that transformational leadership is associated with better teacher-reported outcomes, at least in terms of job satisfaction and organizational commitment [65,66], and school climate (affiliation, innovation, professional interest, and resource adequacy) [67,68] although this was in comparison with transactional or passive-avoidant leadership styles. Moreover, the positive association between transformational leadership and teacher job satisfaction in Malaysia has been inconsistently reported [69]. Finally, it is worthwhile noting that leadership styles in Malaysia are interpreted in the local context [63], and an ideal principal in Malaysia is expected to use multiple leadership styles to match the situation (for example, value-based style during usual circumstances but shift to autocratic leadership during critical times) or the stage of a school’s development [70]. Therefore, school leaders in Malaysia are likely to lead in a way consistent with their values, beliefs, and experience than singly and uniformly adopt the Malaysia Education Blueprint policy-promoted distributed leadership style, which is more akin to the democratic style [64].

## 6. Limitations

It must be noted that the study was conducted using a quantitative method that relies on the respondents to answer according to a measurable scale set; thus, an in-depth perception of the respondent’s answers is not covered in this study. Furthermore, the study focused on the perception of the teachers towards their principals, which cannot be confirmed through documented evidence as the respondents are anonymous. There is a need for further in-depth studies to investigate how far the PLS affect the school culture, effectiveness, and excellence from the perspective of not only the teachers but also the school leaders and stakeholders such as students, the education department, and the community they serve. School leaders should also be aware of the impact of their leadership styles on their teachers and, consequently, the achievement of the school’s vision and mission.

### Concluding Remarks

Teachers’ emotion is important for an effective teaching-learning environment. Principals, as school leaders, must understand the emotional behavior of the teachers they lead and thus guide and instruct them properly. Different principals use different leadership styles, so it was necessary to identify the PLS that best addresses teachers’ emotional behavior. In Malaysia, teachers perceived democratic, instructional, and transformational PLS as the most desired leadership styles in that order. Among these, only the democratic PLS was significantly seen to address the emotional behavior of the teachers. On the other hand, transformational PLS was negatively related, albeit not statistically significant, to address the emotional behavior of the teachers.

Since school leaders determine how well a school operates and contributes significantly to achieving its mission and vision, it is recommended to practice a democratic style to address any emotional issues related to teachers. If school leaders are in tune with the needs of the teachers and have a good rapport that makes teachers feel part of the team, then it is highly likely that the schools’ goals can be achieved.

## Figures and Tables

**Figure 1 behavsci-13-00111-f001:**
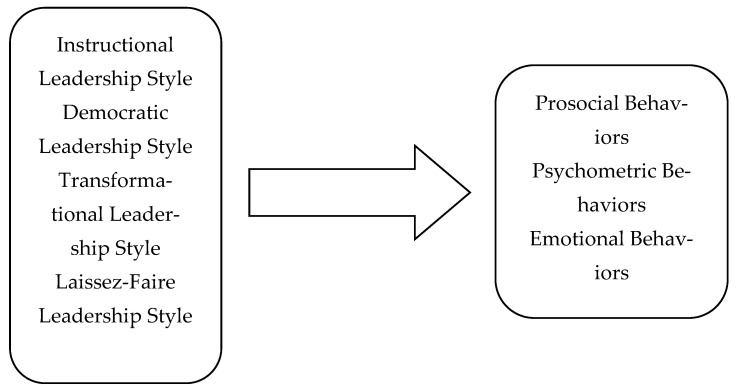
Conceptual framework of the study.

**Table 1 behavsci-13-00111-t001:** Factor loading using EFA. Factor loading for principals’ leadership styles and teachers’ behavior.

No	Item	Factor Loading
	Instructional Leadership style (IL) (α = 0.79)	
IL1	Documented guidelines	0.65
IL2	Share vision and mission	0.82
IL3	Curricular knowledge and instructs teachers	0.60
	Democratic Leadership (DL) (α = 0.81)	
DL1	Fosters friendly relationship	0.88
DL2	Gives opportunity for decision making	0.77
DL3	Supports teacher’s priority needs	0.87
DL4	Encourage teachers to accomplish directed task	0.62
	Transformational Leadership (TL) (α = 0.72)	
TL1	Provides equipment and facilities	0.82
TL2	Open minded in extracting and giving feedbacks	0.74
TL3	Consistently makes changes from the feedbacks	0.83
TL4	Relevant on current technological development and encourage strategic implementation	0.80
	Laissez-faire Leadership (LL) (α = 0.74)	
LL1	Let subordinates work problems out on their own	0.62
LL2	Stays out of the way of subordinates as they do their work	0.71
LL3	As a rule, allows subordinates to appraise their own work	0.73
LL4	Gives subordinates complete freedom to solve problems on their own	0.80
	Teachers’ Emotional Behavior (TE) (α = 0.82)	
TE1	My principal care if I feel left out of things	0.84
TE2	My principal care if I feel lonely	0.79
TE3	My principal care if I feel helpless	0.89
TE4	My principal care if I feel I were someone else	0.91
	Teachers’ Pro-social Behavior (TP) (α = 0.78)	
TP1	My principal appreciates if I often do favors for people without being asked	0.61
TP2	My principal appreciates if I often lend things to people without being asked	0.70
TP3	My principal appreciates if I often help people without being asked	0.82
TP4	My principal appreciates if I often compliment people without being asked	0.81
TP5	My principal appreciates if I often share things with people without being asked	0.81
	Teacher’s psychometric behavior (TPB) (α = 0.76)	
TPB1	My principal cares if I have a headache	0.82
TPB2	My principal cares if I have a stomach ache	0.84
TPB3	My principal cares if I have a back ache	0.84
TPB4	My principal cares if I am feeling low	0.88
TPB5	My principal cares if I am bad tempered	0.64
TPB6	My principal cares if I am feeling nervous	0.58
Eigenvalue (IL, DL, TL, and LF)		15.42, 4.86, 2.20, 1.84
Percentage of variance explained (teachers’ behaviour)		56.78, 13.47, 7.50, 6.21
Eigenvalue (TE, TP, and TPB)		11.29, 3.82, 1.36
Percentage of variance explained (teachers’ behaviour)		64.27, 10.39, 5.86

**Table 2 behavsci-13-00111-t002:** Respondents’ demographic distribution.

Variables	Frequency	Percent
Gender		
Male	40	18.8
Female	173	81.2
Ethnicity		
Malay	168	78.9
Chinese	15	7.0
Indian	23	10.8
Others	7	3.3
Education		
Ph.D.	1	0.5
Master’s degree	82	38.4
Bachelor’s degree	125	58.8
Diploma	5	2.3
Experience		
1–5 years	25	11.8
6–10 years	51	24.0
11–15 years	39	18.2
16 years and above	98	46.0

**Table 3 behavsci-13-00111-t003:** Perception of teachers on principal’s leadership style.

Items	Mean ± SD	Perception
Instructional Leadership		
Item 1: documented guidelines	4.25 ± 0.522	high
Item 2: share vision and mission	4.28 ± 0.432	high
Item 3: curricular knowledge and instructs teachers	4.70 ± 0.721	high
Democratic Leadership		
Item 1: fosters friendly relationship	4.32 ± 0.412	high
Item 2: gives opportunity for decision making	4.72 ± 0.701	high
Item 3: supports teacher’s priority needs	4.18 ± 0.641	high
Item 4: encourage teachers to accomplish directed task	4.22 ± 0.373	high
Transformational Leadership		
Item 1: provides equipment and facilities	4.12 ± 0.711	high
Item 2: open-minded in extracting and giving feedback	4.78 ± 0.515	high
Item 3: consistently makes changes from the feedback	4.68 ± 0.246	high
Item 4: relevant on current technological development and encourage strategic implementation	4.53 ± 0.321	high
Laissez-faire Leadership		
Item 1: let subordinates work problems out on their own	3.80 ± 0.733	average
Item 2: stays out of the way of subordinates as they do their work	3.23 ± 0.792	average
Item 3: as a rule, allows subordinates to appraise their own work	3.12 ± 1.141	average
Item 4: gives subordinates complete freedom to solve problems on their own	3.87 ± 1.013	average

**Table 4 behavsci-13-00111-t004:** Comparison of teachers’ preferred leadership styles.

Leadership Style	*n*	Mean ± SD [Range]
*Instructional*	213	4.41 ± 0.56 [3.00–5.00]
*Democratic*	213	4.36 ± 0.53 [2.75–5.00]
*Transformational*	213	4.53 ± 0.55 [2.50–5.00]
*Laissez-faire*	213	3.50 ± 0.92 [1.75–5.00]

**Table 5 behavsci-13-00111-t005:** Perception of teachers on the principal’s handling of teachers’ needs.

Items	Mean ± SD	Perception
Emotional Problem		
Item 1: My principal care if I feel left out of things	3.76 ± 0.711	average
Item 2: My principal care if I feel lonely	3.67 ± 0.904	average
Item 3: My principal care if I feel helpless	3.81 ± 0.366	average
Item 4: My principal care if I feel I were someone else	3.73 ± 0.815	average
Pro-social Behavior		
Item 1: My principal appreciates if I often do favors for people without being asked	3.78 ± 0.746	average
Item 2: My principal appreciates if I often lend things to people without being asked	3.71 ± 0.927	average
Item 3: My principal appreciates if I often help people without being asked	4.12 ± 0.836	high
Item 4: My principal appreciates if I often compliment people without being asked	4.22 ± 0.656	high
Item 5: My principal appreciates if I often share things with people without being asked	4.20 ± 0.615	high
Psychometric		
Item 1: My Principal cares if I have a headache	3.64 ± 0.718	average
Item 2: My Principal cares if I have a stomach-ache	3.80 ± 0.710	average
Item 3: My Principal cares if I have a backache	3.65 ± 0.705	average
Item 4: My Principal cares if I am feeling low (depressed)	3.83 ± 0.760	average
Item 5: My Principal cares if I am irritable or bad-tempered	3.45 ± 0.815	average
Item 6: My Principal cares if I am feeling nervous	3.51 ± 0.757	average
Item 7: My Principal cares if I have difficulties falling asleep	3.72 ± 0.838	average
Item 8: My Principal cares if I am feeling dizzy	3.62 ± 0.947	average

**Table 6 behavsci-13-00111-t006:** Multiple linear regression: association between leadership style and teachers’ needs.

Dependent Variables	R	R Square	Adjusted R Square	Std. Error of the Estimate	Change Statistics				
					R Square Change	F Change	df1	df2	Sig. F Change
Emotional	0.264 ^a^	0.069	0.035	0.85534	0.064	1.712	5	207	0.244
Pro-social	0.302 ^a^	0.091	0.062	0.78088	0.091	2.584	5	207	0.023
Psychometric	0.355 ^a^	0.126	0.045	0.81803	0.082	2.163	5	207	0.081

^a^ Predictors: laissez-faire leadership, instructional leadership, democratic leadership, and transformational leadership.

**Table 7 behavsci-13-00111-t007:** ANOVA: Association between leadership style and teachers’-related variables.

Dependent Variable	Model		Sum of Squares	df	Mean Square	F	Sig.
Emotional	1	Regression	3.882	5	0.887	1.712	0.244 ^a^
		Residual	62.554	207	0.587	-	-
		Total	66.436	212	-	-	-
Pro-social	1	Regression	4.884	5	1.283	2.584	0.023 ^a^
		Residual	52.823	207	0.368	-	-
		Total	57.707	212	-	-	-
Psychometric	1	Regression	5.491	5	1.456	2.163	0.081 ^a^
		Residual	69.819	207	0.544	-	-
		Total	75.310	212	-	-	-

^a^ Predictors: laissez-faire leadership, instructional leadership, democratic leadership, and transformational leadership.

## Data Availability

Data is stored with the corresponding author. It will be available upon request.

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
