# Peer review of "Relationship between Principals’ Leadership Styles and Teachers’ Behavior"

_behavsci, 2023, doi:10.3390/bs13020111_

Round 1

Reviewer 1 Report

Thank you for the opportunity to review this paper for Behavioral Sciences

Aim

The aim of this paper is to analyse how a school principal’s leadership style affects teachers’ emotional, pro-social and psychometric needs. Three research questions guided the study.

Literature 

To address the central issue of the study, the author has conducted a review of the relevant literature about leadership and the conceptual framework that involved human emotion needs, psychological needs, and Herzberg’s two-factor theory. The text was logically presented.

Methodology

The research used a survey technique that involved 258 teachers, 213 of whom were included in the analysis. Data were collected form instruments that were sufficiently described, and the data analysis procedures were clear to follow.

Results and Discussion

The results are presented clearly in terms of the data collected and analysed. However, the research results would be better presented in terms of the research questions. See other comments under Critique.

The discussion is consistent with the data and is examined in the context of previous related research. That there were no statistically significant relationships identified from this study between PLS and the teachers’ needs, it may be helpful to decide what additional or different data might be needed to answer the last research question.

Critique

This is a potentially useful paper with the following issues needing to be addressed in the revision.

1.     The research design was not overtly stated. Please explain the research design in terms of accepted design in educational research.

2.     Table 1 about normality need not be a part of the paper – could be omitted or place din an Appendix. To understand the normality analysis, the text is sufficient for the reader.

3.     It is not clear how Herzberg’s two-factor theory was used to analyse the data and there is no mention in the discussion. I recommend omitting Herzberg’s two-factor theory. The emotional needs and psychological needs are sufficient to analyse and explain the data and the findings.

4.     The research results would be better presented in terms of the research questions. Then three of the research questions present useful information

5.     The items for each of the instruments are presented in groups in Tables 3 and 5. However no evidence has been provided that these items fit in these groups. I am looking for factor analyses. Can the authors check that these items fit each topic/scale?

6.     In Table 7, the significant column has without any explanation what this means. 

7.     The data in Table 6 and related text show that PLS was not associated with the teachers’ needs. Consequently, there is no reason to report the data in Table 8 – so Table 8 should be omitted as it provides no information that is useful from this research. The related statement is sufficient.

8.     For the references, sometimes the article is in title case other times in lower case. Be consistent with the requirement of the journal. 

Author Response

I have attached the responses along with the revised version of the manuscript

Reviewer 2 Report

Introduction and theoretical framework:

A good theoretical framework is presented with many topical and relevant references in the field.

Material and methods:

Appropriate statistical methods are used, as well as the use of validated instruments which allows giving robustness to the results as well as to the conclusions drawn from them.

Results, discussion and conclusions: They are presented in a correct and orderly manner, which facilitates reading and interpretation by the reader. In addition, the fact that they are broken down into different groups allows for an in-depth analysis of the different variables involved.

General: This is a relevant topic, since leadership in educational centers directly affects their quality as well as the performance of their workers, so these studies are very necessary, as well as their possible replication in other contexts.

Author Response

I have attached here the authors' responses with revised manuscript..

Round 2

Reviewer 1 Report

Thank for attending to the issues raised.

The paper now reads in a more logical and coherent manner.

I have no further issues to raise.